# The Antibacterial Potential of Brazilian Red Propolis against the Formation and Eradication of Biofilm of *Helicobacter pylori*

**DOI:** 10.3390/antibiotics13080719

**Published:** 2024-08-01

**Authors:** Mariana B. Santiago, Matheus H. Tanimoto, Maria Anita L. V. Ambrosio, Rodrigo Cassio S. Veneziani, Jairo K. Bastos, Robinson Sabino-Silva, Carlos Henrique G. Martins

**Affiliations:** 1Laboratory of Antimicrobial Testing, Institute of Biomedical Sciences, Federal University of Uberlândia, Uberlândia 38405-320, Brazil; mariana.brentini@ufu.br; 2Faculty of Pharmaceutical Sciences of Ribeirão Preto, University of São Paulo, Ribeirão Preto 14040-900, Brazil; tanimoto@usp.br (M.H.T.); jkbastos@fcfrp.usp.br (J.K.B.); 3Nucleus of Research in Sciences and Technology, University of Franca, Franca 14404-600, Brazil; malvasconcelos@yahoo.com.br (M.A.L.V.A.); rodrigo.veneziani@unifran.edu.br (R.C.S.V.); 4Innovation Center in Salivary Diagnostic and Nanobiotechnology, Department of Physiology, Institute of Biomedical Sciences, Federal University of Uberlandia, Uberlândia 38408-100, Brazil; robinsonsabino@gmail.com

**Keywords:** *Helicobacter pylori*, Brazilian red propolis, natural products, antibiofilm, time–kill, nucleotide leakage

## Abstract

*Helicobacter pylori* is associated with gastrointestinal diseases, and its treatment is challenging due to antibiotic-resistant strains, necessitating alternative therapies. Brazilian red propolis (BRP), known for its diverse bioactive compounds with pharmaceutical properties, was investigated for its anti-*H. pylori* activity, focusing on biofilm formation inhibition and eradication. BRP was tested against *H. pylori* (ATCC 43526) using several assays: time–kill, nucleotide leakage, biofilm formation inhibition (determining the minimum inhibitory concentration of biofilm of 50%—MICB_50_, and cell viability), and biofilm eradication (determining the minimum eradication concentration of biofilm of 99.9%—MBEC). Standardization of *H. pylori* biofilm formation was also conducted. In the time–kill assay, BRP at 50 µg/mL eliminated all *H. pylori* cells after 24 h. The nucleotide leakage assay showed no significant differences between control groups and BRP-treated groups at 25 µg/mL and 50 µg/mL. *H. pylori* formed biofilms in vitro at 10^9^ CFU/mL after 72 h. The MICB_50_ of BRP was 15.6 µg/mL, and at 500, 1000, and 2000 µg/mL, BRP eradicated all bacterial cells. The MBEC was 2000 µg/mL. These findings suggest that BRP has promising anti-*H. pylori* activity, effectively inhibiting and eradicating biofilms. Further studies are necessary to elucidate BRP’s mechanisms of action against *H. pylori*.

## 1. Introduction

*Helicobacter pylori*, a Gram-negative, flagellated, microaerophilic bacillus, colonizes the gastric mucosa [1,2,3]. It is linked to chronic gastric diseases, including gastritis and gastric and duodenal ulcers, and increases gastric cancer risk [1,3,4]. *H. pylori* is highly prevalent, colonizing over half the global population [5]. Transmission likely occurs in childhood via oral–oral and fecal–oral routes. Many carriers remain asymptomatic or do not develop pathologies [6,7]. However, eradicating *H. pylori* in symptomatic individuals is difficult. The standard treatment is triple therapy: a proton pump inhibitor (e.g., omeprazole) and two antibiotics (e.g., clarithromycin, tetracycline, amoxicillin) [3]. High failure rates are due to virulence factors that foster antibiotic resistance [2,3,8]. These factors (e.g., motility, toxin production, urease activity, outer membrane proteins) enable *H. pylori* to colonize the hostile gastric environment [2,7]. The capability of biofilm formation is a virulence factor present in *H. pylori*, which is one of the primary factors associated with the development of antibiotic resistance [2,9,10,11].

The World Health Organization (WHO) recognizes *H. pylori*’s antibiotic resistance as a serious health issue. In 2017, WHO listed 12 antibiotic-resistant bacteria threatening human health, prioritizing the development of new agents to eradicate clarithromycin-resistant *H. pylori* [12]. However, even though there is a demand for new antibiotics capable of eliminating bacteria resistant to current antibiotics, and acknowledging that the biofilm-forming ability of the bacterium plays a significant role in the development of this resistance, the biofilm-forming ability of *H. pylori* is still under-explored in scientific research. On the other hand, studies on therapeutic options with antibiofilm properties have shown promising results, primarily involving natural products or their derivatives [2].

Natural products originate from sources like plants, animals, minerals, and microorganisms. They are scientifically important due to their unique active substances with biological properties [13]. Their use in healthcare is ancient, embodying folk knowledge passed down through generations [14]. Propolis is a natural product produced by bees, commonly known as “bee glue”. It is formed through a combination of buds, flowers, and various plant exudates with bee saliva secretions, wax, and pollen [15] and is composed mainly of resin (50%), wax (30%), essential oils (10%), pollen (5%), and organic compounds (5%). Its composition can vary by geographic location and collection time due to the seasonality of natural products [16,17]. As a consequence, approximately 500 different chemical components have been identified in propolis from various sources and locations. Among these, significant compounds have been identified in propolis, such as flavonoids, phenolic compounds, alcohols, and terpenes [18]. In Brazil, red propolis is the result of bees, *Apis mellifera*, collecting red exudates from the surface of the trunks of *Dalbergia ecastophyllum*, commonly known in Brazil as “rabo de bugio” [19,20,21].

Owing to its botanical source and the vast biodiversity of Brazil, Brazilian red propolis (BRP) contains compounds not found in propolis from other sources [21]. BRP has had many of its chemical compounds identified, such as flavonoids (liquiritigenin), chalcones (isoliquiritigenin), pterocarpans (medicarpin), isoflavones (formononetin), isoflavanes (vestitol, neovestitol, and 7-O-methylvestitol), and prenylated benzophenones (a mixture of guttiferone E/xanthochymol and oblongifolin A) [22,23,24]. From a chemotaxonomic perspective, these constituents found in BRP, especially the isoflavonoids, are chemotaxonomic markers characteristic of members of the subfamily Papilionoideae (Fabaceae), to which the genus *Dalbergia* belongs [25]. The literature reveals various biological properties attributed to BRP, such as antibacterial, antiparasitic, and antioxidant, among others [20]. Two studies have already reported promising anti-*H. pylori* activity of BRP against the sessile form of the bacteria [26,27].

Therefore, considering that bacterial resistance is highly associated with the ability of bacterial biofilm formation [2], that natural products have already shown promising antibiofilm activity against *H. pylori* [2], that there is an urgent need for the development of new therapeutic options against *H. pylori* [12], and that studies have already demonstrated promising anti-*H. pylori* activity of BRP against free-living forms of the bacteria [26,27], it is encouraged to deepen this evaluation to determine if BRP also exhibits antibiofilm activity against *H. pylori*. For this purpose, the current study evaluated in vitro the potential of the crude hydroalcoholic extract of BRP against the formation and eradication of the biofilm of *H. pylori* (ATCC 43526).

## 2. Results

### 2.1. Chemical Characterization

The chromatographic profile of BRP (Figure 1) identified the components liquiritigenin, formononetin, vestitol, neovestitol, medicarpin, 7-O-neovestitol, guttiferone E, and oblongifolin B.

### 2.2. Time–Kill

The time–kill curve of *H. pylori* (ATCC 43526) is represented in Figure 2. It was determined that after 24 h of exposure of the bacteria to 50 µg/mL of BRP, the colony forming unit (CFU) count was reduced to zero. When exposed to 25 µg/mL of BRP, even after 72 h, it did not eliminate bacterial growth. Tetracycline at 0.74 µg/mL, used as a positive control, eliminated the growth within 56 h of exposure.

### 2.3. Nucleotide Leakage

The results obtained in the nucleotide leakage assay are represented in Figure 3 and demonstrate that BRP does not cause damage to the cell membrane of *H. pylori* (ATCC 43526), as no statistical differences were observed between the optical density (OD) obtained with the treatments of 25 and 50 µg/mL of BRP compared to the OD obtained in the control (bacteria only).

### 2.4. Biofilm Formation

In the assay for the standardization of biofilm formation, it was observed that after 3 days of incubation at an inoculum concentration of 10^9^ CFU/mL (Figure 4), *H. pylori* (ATCC 43526) formed a biofilm in satisfactory quantities for the continuation of the experiments. Therefore, this inoculum concentration and incubation time were selected for the continuation of the experiments on the inhibition of biofilm formation and biofilm eradication.

### 2.5. Inhibition of Biofilm Formation

The BRP was able to inhibit 50% of the biofilm formation of the bacteria at a concentration of 15.6 µg/mL (minimum inhibitory concentration of biofilm (MICB_50_) value). Additionally, it was observed that at concentrations of 500, 1000, and 2000 µg/mL, there was no cell growth (Figure 5a). Tetracycline achieved the same action at a concentration of 0.0461 µg/mL, but was not able to eliminate viable cells at the evaluated concentrations (Figure 5b).

### 2.6. Biofilm Eradication

The data from the eradication assay are represented in Figure 6. It was determined that the minimum biofilm eradication concentration (MEBC) of BRP is 2000 µg/mL. Furthermore, it is noticeable that in addition to eliminating 99.9% of the cells, BRP also achieved lower OD values, which represent the biofilm aggregate, compared to the control (Figure 6a). On the other hand, tetracycline (Figure 6b) also obtained lower OD values than the control; however, at the evaluated concentrations, it was not possible to determine an MEBC value, as cells were still viable at the highest concentration evaluated.

## 3. Discussion

To the best of our knowledge, the results obtained in the present study regarding the properties of BRP against the bacteria *H. pylori* are unprecedented in the scientific literature. Although there are some studies that have evaluated the antibacterial properties of propolis samples from different locations [28,29,30,31], their antibiofilm property has not been explored. In previous studies, our research group determined that the minimum bactericidal concentration (MBC) of BRP against *H. pylori* (ATCC 43526) is 50 µg/mL [27]. Due to this, this concentration was chosen for evaluating the *H. pylori* time–kill curve, with the BRP concentrations assessed being 50 µg/mL (1× MBC) and 25 µg/mL (1/2 MBC). Tetracycline was also used at the MBC concentration defined by the authors (0.74 µg/mL) [27].

Therefore, our study determined that the time required for BRP at a concentration of 50 µg/mL to completely eliminate *H. pylori* (ATCC 43523) is 24 h, and its sub-bactericidal concentration (25 µg/mL) was not able to eliminate the bacteria even after 72 h of exposure. The antibiotic tetracycline at a concentration of 0.74 µg/mL took 56 h to eliminate the bacteria, which is 32 h more than what was necessary for BRP at 50 µg/mL. Due to the lack of studies that have charted this time–kill curve profile for propolis samples, it is not possible to discuss the results of the present study with other authors. However, comparing with other natural products, Ye, et al. [32] used *Chenopodium ambrosioides* to determine the time–kill of *H. pylori* (NCTC 11637), and complete elimination of the bacteria occurred in 24 h at concentrations of 1x and 2x the determined minimum inhibitory concentration (16,000 µg/mL). Taking together the data from the present study and that of the authors [32], it can be suggested that natural products may exhibit bactericidal activity against *H. pylori* within the first 24 h of contact.

Some antibacterial agents have the property of causing irreversible damage to the bacterial cytoplasmic membrane. When the leakage of cytoplasmic content is detected, this release of intracellular content is used as an indicator of membrane integrity [33]. In the present study, no statistical differences were observed in the nucleotide leakage between the control and those treated with BRP at 25 and 50 µg/mL after 24 h of exposure, indicating that the bacterial death caused by BRP does not cause any damage to the membrane of *H. pylori* (ATCC 43526).

Another novel finding in the scientific literature presented by this study is about the antibiofilm property of BRP against *H. pylori* (ATCC 43526). No other studies were found that evaluated the antibiofilm action, even though there have been studies demonstrating the antibacterial potential of propolis in both in vitro and in vivo assays [26,27]. The biofilm-forming capability of *H. pylori* is one of the mechanisms associated with the development of antibiotic resistance, with the particularity being that the bacteria can change its classic bacillus morphology to coccoid-shaped, thus obtaining a more resistant biofilm [11].

In the present study, the *H. pylori* strain (ATCC 43526) proved capable of forming biofilm at a concentration of 10^9^ after 72 h of incubation at 37 °C in a CO_2_ incubator. The biofilm formation inhibition assay determined that BRP can inhibit biofilm growth by 50% at a concentration of 15.6 µg/mL and completely eliminate viable cells at concentrations starting from 500 µg/mL. Due to the complexity of the bacterial community in biofilm, it is estimated that the dose of an antibacterial agent required to effectively eliminate the biofilm may be 10- to 1000-fold higher than the dose needed to eliminate the bacteria in its planktonic form [34,35,36]. The current study corroborates this estimate, as the necessary concentration of BRP to completely eliminate the biofilm of *H. pylori* (ATCC 43526) was 500.0 µg/mL, and in other studies a BRP concentration of 50.0 µg/mL was sufficient to eliminate the planktonic cells of the same bacteria.

Sa Assis, et al. [37] conducted a study evaluating the potential of the aqueous extract of Brazilian green propolis in inhibiting the formation of biofilm by bacteria associated with endodontic infections. The bacteria tested included *Fusobacterium nucleatum* (ATCC 25586), *Parvimonas micra* (ATCC 23195), *Prevotella intermedia* (ATCC 35406), *Porphyromonas gingivalis* (ATCC 33277), and *P. endodontalis* (ATCC 33563). The authors determined that the extract was effective in reducing both the biomass and the cellular viability of the evaluated bacteria at a concentration of 110,000.0 µg/mL, with the exception of *P. micra*, for which the most effective concentration for reducing viability was 55,000.0 µg/mL [37].

Other researchers, Meccatti, et al. [38], also investigated the inhibitory action on biofilm formation of Brazilian green propolis. They assessed the effectiveness of a combination of hydroethanolic extract of Brazilian green propolis (concentration of 5000.0 µg/mL) with *Cinnamomum verum* (concentration of 2200.0 µg/mL) against various strains of *Acinetobacter baumannii* and *Pseudomonas aeruginosa*. The authors found that, in some cases, the combination of these two natural products was statistically more effective in inhibiting biofilm formation than the use of chlorhexidine, which was used as a positive control in the study [38].

The studies mentioned above by Sa Assis, et al. [37] and Meccatti, et al. [38], regarding the antibiofilm action of Brazilian green propolis extracts against different bacterial species, combined with the results obtained in the present study on the antibiofilm property of BRP against *H. pylori* (ATCC 43526), highlight the biological potential of these extracts and how Brazilian biodiversity is capable of producing resources with active potential against various pathogenic bacteria.

When it comes to antibiofilm action, there are various opinions about which strategy is more relevant. Wei, et al. [39] argue that preventing biofilm formation is more effective, as it prevents the biofilm from becoming mature. On the other hand, Olson, et al. [40] contend that the efficacy of an agent in eliminating mature biofilm increases the success rate in the treatment of clinical diseases. In the present study, BRP showed promise against *H. pylori* (ATCC 43526) in both strategies, as it was capable of completely inhibiting the cellular viability of biofilm formation at a concentration of 500.0 µg/mL, and also of completely eliminating the viability of mature biofilm in the eradication assay at a concentration of 2000.0 µg/mL.

The study by Meccatti, et al. [38], previously mentioned, which assessed the activity of Brazilian green propolis in combination with *C. verum* extract against the bacteria *A. baumannii* and *P. aeruginosa*, also evaluated the effect of this combination on mature biofilms of the same bacteria. In all the assays conducted, the authors determined that there was a significant reduction in mature biofilm density when the concentration of green propolis extract in the mixture was 5000.0 µg/mL [38]. In the current study, BRP achieved total eradication of mature *H. pylori* (ATCC 43526) biofilm at a concentration of 2000 µg/mL. Once again, the data obtained here, as well as by other authors [38], highlight that natural products from Brazilian sources are a source of bioactive compounds of great interest in combating pathogenic bacteria and their mechanisms of action.

Regarding the chemical components identified in the BRP used in the present study, only two compounds have been investigated for their anti-*H. pylori* activity. Fukai, et al. [41] evaluated the compounds vestitol and formononetin. The authors found that vestitol exhibited anti-*H. pylori* activity against strains resistant to clarithromycin and amoxicillin, while formononetin was considered by the authors to have weak activity. Studies related to the anti-*H. pylori* activity of the compounds liquiritigenin, neovestitol, medicarpin, 7-O-neovestitol, guttiferone E, and oblongifolin B have not yet been reported in the scientific literature. However, there are studies suggesting that the compounds medicarpin, vestitol, and neovestitol may be responsible for the antibacterial activity exhibited by BRP [42,43]. Additionally, other studies debate that, due to different compounds exhibiting antibacterial activity against different strains, this property may be related to a possible synergistic effect occurring between the components of the crude extract of BRP [42,44].

As previously mentioned, the region of collection or seasonality can alter the composition of natural products. There are differences in the compounds identified between Cuban red propolis and BRP, even though both have the *Dalbergia* genus as their botanical source [45,46], highlighting how the region can influence the chemical composition of a natural product. Regarding BRP, some studies have already analyzed the effects of seasons on the chemical composition of BRP from various regions of Brazil, such as Bahia [22], Alagoas [24], and Pernambuco [47]. In all data, it is noted that there is no qualitative change in the identified compounds across the seasons; rather, a quantitative change in the components occurs. That is, even in different regions and seasons, BRP consistently presents the same compounds, with only the quantity varying depending on the season. These data were taken into consideration in the decision not to use more than one BRP sample from different regions in the present study, as the literature already reports the same components in their chemical compositions, which are attributed to the biological properties of natural products [24].

Another point regarding the present study that needs to be mentioned is the choice to use only one reference strain. The strain used, *H. pylori* (ATCC 43526), is part of a collection and was isolated from the gastric antrum, the site of bacterial infection, making it an ideal model for the objectives of this work. As this is the first report evaluating the anti-biofilm properties of BRP against *H. pylori*, it is essential that the data presented allow for reproducibility [48]. Nonetheless, it is also important to acknowledge that reference strains may not exhibit the same characteristics as clinical isolates [48]. Therefore, further investigation into the effects of BRP against *H. pylori* is necessary before it can truly become an alternative therapy in combating the bacteria, such as studies including a wider variety of strains, including antibiotic-resistant clinical isolates, and analyzing the anti-*H. pylori* properties of isolated BRP compounds. However, the results presented in this study contribute further to the advancement of this topic and bring unprecedented novelties.

## 4. Materials and Methods

### 4.1. Chemical Characterization

#### 4.1.1. Acquisition and Extraction of Red Propolis

The red propolis sample was sourced from the Cooperative of Beekeepers of Canavieiras (COAPER) in Canavieiras, BA, Brazil in August 2022, and was stored in a freezer at −26 °C. This study was registered with the National System for Management of Genetic Heritage and Associated Traditional Knowledge (SisGen) under the registration number AF234D8. One kilogram of red propolis was placed in Erlenmeyer flasks and mixed with a hydroalcoholic solution (70% ethanol and 30% water). After two hours of immersion, the propolis was ground using a mixer-type processor, and the extractor solvent volume was adjusted to a 1:5 (g/mL) ratio. The flasks were kept in a shaker-type incubator (INNOVA 4300, New Brunswick Scientific, Saint Albans, UK) at 120 rpm and 37 °C for 24 h. The mixture was then filtered using filter paper. The extraction process was repeated three times. After filtration, the solvent was evaporated in a rotary evaporator below 40 °C. The resulting extract was lyophilized and analyzed using high-performance liquid chromatography with a diode array detector (HPLC-DAD) to determine its chromatographic profile.

#### 4.1.2. Extract Analysis by HPLC-DAD

In this study, we ensured the reliability of our results by analyzing the crude extract using HPLC-DAD, a method established in previous research by the group [22,25]. The extract was dissolved in HPLC-grade methanol at a concentration of 1 mg/mL in 1.5 mL microtubes, filtered through polytetrafluoroethylene (PTFE) filters (0.45 µm), and stored in vials for HPLC-DAD analysis.

The analysis was conducted using a Waters^®^ 2695 HPLC system (Milford, CT, USA) equipped with binary pumps (model 1525), an automatic injector (model 2707), and a diode array detector (model 2998) interfaced with a computer running Empower 3 software. The stationary phase utilized a Supelco Ascentis Express C18 column (2.7 μm × 150 mm × 4.6 mm) coupled with an analytical Synergi Polar-RP pre-column (4.0 × 3.0 mm, 4 μm).

The mobile phase consisted of purified water (Millipore, MA, EUA) acidified with formic acid (0.1%; Synth^®^, Diadema, Brazil) (A) and HPLC-grade acetonitrile (SK Chemicals, Seongnam, Republic of Korea) (B), with a flow rate of 1 mL/min and an injection volume of 10 µL. The elution was gradient-based, starting with 20% B, increasing from 20% to 50% B over 40 min, then to 100% B by 90 min, holding at 100% B until 95 min, and finally returning to 20% B, with a total run time of 100 min. Detection was carried out over a wavelength range of 210–600 nm, with chromatograms monitored between 240 and 315 nm.

### 4.2. Anti-H. pylori Activity

#### 4.2.1. Bacteria Used in the Assays

The *H. pylori* strain used was sourced from the American Type Culture Collection (ATCC 43526, Manassas, VA, USA) and is part of the culture collection of the Laboratory of Antimicrobial Testing (LEA) at the Federal University of Uberlândia (UFU). The strain was preserved under cryopreservation at −80 °C in Brucella broth (Difco Labs, Detroit, MI, USA) supplemented with 10.0% (*v*/*v*) bovine fetal serum and containing 20.0% (*v*/*v*) glycerol. For bacterial cultivation, *H. pylori* was seeded on Brucella agar (Difco) containing 5% of defibrinated horse blood or in Brucella broth (Difco) supplemented with 10.0% bovine fetal serum. It was then incubated in a CO_2_ incubator (Panasonic Biomedical, Amsterdam, The Netherlands) at 37 °C for 72 h in an atmosphere containing 10.0% CO_2_.

#### 4.2.2. Time–Kill

The time–kill assay of BRP against *H. pylori* (ATCC 43526) was conducted in triplicate, as described by Sposito, et al. [49], with minor modifications. For the assay, the bacterial inoculum was adjusted to contain 5 × 10^5^ CFU/mL. BRP was added at concentrations of 25 and 50 µg/mL, and tetracycline, evaluated at a concentration of 0.74 µg/mL, was used as a positive control. The bacterial inoculum without any treatment was used as a negative control. The assay was incubated in a CO_2_ incubator at 37 °C. To perform the viable bacterial colony count, aliquots were removed at 0 and 30 min, and at 2, 4, 8, 12, 24, 32, 36, 48, 56, 60, and 72 h. The aliquots were diluted in a 1:10,000 ratio and seeded on Brucella agar containing 5% defibrinated horse blood. The agar was incubated under the necessary conditions, and after 72 h, colony counting was conducted. The results are expressed as mean ± standard deviation (SD). The time–kill curve was constructed by plotting log_10_ CFU/mL against time using GraphPad Prism (version 8.0) software (GraphPad Software, Boston, MA, USA).

#### 4.2.3. Nucleotide Leakage

To evaluate whether BRP causes damage to the cell membrane of *H. pylori*, a nucleotide leakage assay was performed. The assay was conducted as described by Chen and Cooper [33], with modifications. Briefly, the bacterial inoculum was adjusted to achieve a final concentration of 5 × 10^5^ CFU/mL in three test tubes, one containing 25 µg/mL of BRP, another 50 µg/mL of BRP, and the third containing only the inoculum. Additionally, three tubes containing only broth and BRP at the evaluated concentrations were used as blanks. The tubes were incubated in a CO_2_ atmosphere for 24 h. After incubation, the contents of the tubes were filtered through 0.2 µm syringe filters, and the absorbance was measured at 260 and 280 nm using a microtiter plate reader (GloMax^®^, Promega, Madison, WI, USA). The absorbance values obtained were subtracted from their respective blanks by subtraction. The assay was performed in triplicate, and the results of optical density (OD) were graphically represented using GraphPad Prism (version 8.0) software (GraphPad Software).

#### 4.2.4. Standardization of Biofilm Formation

The optimal inoculum concentration and incubation time for biofilm formation were determined using the biofilm formation standardization assay, adapting the Sandberg, et al. [50] methodology for *H. pylori* in order to respect the nutritional and physiological characteristics of the bacteria. The bacteria were incubated in a CO_2_ incubator for 3, 5, and 7 days at concentrations ranging from 10^6^ to 10^9^ CFU/mL in 96-well microplates. After incubation, the contents were removed from all wells, and they were washed with Milli-Q water to ensure removal of planktonic cells. The adhered biofilm in the microplate was fixed with methanol for 20 min, after which the methanol was removed, and crystal violet (0.2% *v*/*v*) was added to the wells for 10 min. The crystal violet was then removed and excess washed off with tap water. After drying, 200 µL of 33% (*v*/*v*) acetic acid was added to the wells to solubilize the dye adhered to the cells. After 30 min, the reading was taken at 595 nm using a microtiter plate reader (GloMax^®^). The assay was carried out in triplicate, and the OD values obtained were expressed graphically.

#### 4.2.5. Inhibition of Biofilm Formation

To evaluate the potential of BRP to inhibit biofilm formation of *H. pylori*, the study assessed the minimum inhibitory concentration of biofilm (MICB_50_), which is the lowest concentration of the sample evaluated capable of inhibiting the formation of 50% or more of the biofilm [39]. Additionally, the assessment of cell viability was conducted through CFU count. Two 96-well microplates were prepared identically for the assessment. Briefly, dilutions of BRP were made, with final concentrations ranging from 0.98 to 2000 µg/mL. Tetracycline was used as a positive control, with concentrations ranging from 0.0115 to 5.9 µg/mL. An inoculum of *H. pylori* at a concentration of 10^9^ CFU/mL was added to all wells, and a control with only the inoculum and Brucella broth was also used. The microplates were incubated for 72 h at 37 °C in a CO_2_ incubator.

After incubation, the microplate designated for determining the MICB_50_ underwent the process described above, standardized by Sandberg, et al. [50], for quantification of OD. The percentage of inhibition was calculated by using the equation:$$1-\left(\frac{{\mathrm{At}}_{595}\;\mathrm{nm}}{{\mathrm{Ac}}_{595}\;\mathrm{nm}}\right)\times 100$$
where At_595_ nm and Ac_595_ nm are the absorbance values of the wells treated with the samples and the control, respectively [39].

The microplate designated for CFU counting had the contents of its wells aspirated, and was carefully washed with Milli-Q water to eliminate residual planktonic cells. After this, new broth was added to all wells and the microplate underwent a sonication process so that the adhered cells could be dislodged by vibration. Dilutions ranging from 10^−1^ to 10^−7^ were performed per well, and 50 µL of each dilution were plated on new Brucella agar containing 5.0% defibrinated horse blood, which was then incubated according to the needs of the bacteria. After incubation, the colonies were counted and the results expressed in log_10_ CFU/mL. All assays were performed in triplicate and the results were graphically represented.

#### 4.2.6. Biofilm Eradication

To determine if BRP has the ability to eradicate *H. pylori* biofilm, the minimum biofilm eradication concentration (MBEC) was assessed. This is defined as the lowest concentration of the sample capable of reducing the number of viable cells in the biofilm by at least 99.9% [51]. Additionally, the biomass was evaluated by obtaining the OD [50].

The assay to evaluate the eradication of *H. pylori* biofilm by BRP is similar to the one used for assessing biofilm formation inhibition, with the main difference being in the initial steps. Initially, only the bacterial inoculum (10^9^ CFU/mL) is added to all wells of the microplates. This is then left to form and adhere as a biofilm to the microplate at 37 ºC for 72 h in a CO_2_ incubator. After this period, the contents of the wells are removed, and fresh broth containing varying concentrations of BRP (0.98 to 2000 µg/mL), tetracycline (0.0115 to 5.9 µg/mL) for testing the eradication potential of these substances on the established biofilm, or it contains only the broth (serving as a negative control, with the inoculum only) is carefully added to the wells. The microplates are then reincubated under the same conditions for an additional 72 h.

After the incubation period, one microplate underwent the process previously described using crystal violet as a dye to obtain the OD [50]. The other microplate underwent the same process, also previously described, to perform the count of CFU/mL. MEBC value was calculated by dividing the count of microorganisms obtained in the treatments by the count obtained in the negative control (inoculum only) and multiplying the result by 100 to obtain the percentage of biofilm eradication [51]. All assays were performed in triplicate, and the results were graphically expressed.

#### 4.2.7. Statistical Analysis

The results of the nucleotide leakage assay are presented as means ± standard deviations. The normality of the data was assessed using the Shapiro–Wilk test using GraphPad Prism (version 8.0) software (GraphPad Software). Since the data showed a normal distribution, it was then analyzed using one-way analysis of variance (ANOVA). Statistical differences were considered significant at *p* < 0.05.

## 5. Conclusions

Based on the data obtained in this study, it can be concluded that BRP has promising antibiofilm activity against *H. pylori* (ATCC 43526) under the evaluated in vitro conditions, as it was capable of completely eliminating bacterial cells in both biofilm formation and eradication, and was able to inhibit 50% of biofilm formation. It is also determined that the time required for BRP to eliminate planktonic cells of *H. pylori* is 24 h. Furthermore, it can be suggested that the bacterial death of the planktonic cells does not cause damage to the cytoplasmic membrane, as indicated by the nucleotide leakage assay. This study emphasizes the importance of investigating and exploring natural products as sources of innovative therapeutic solutions. The perspectives outlined by these findings highlight the ongoing need for interdisciplinary research that integrates biology, chemistry, and medicine to unravel the secrets of bioactive compounds and their applications. As research progresses, the hope is that these studies will lead not only to more effective therapeutic approaches but also inspire a renewed commitment to biodiversity conservation and the promotion of human health.

## Figures and Tables

**Figure 1 antibiotics-13-00719-f001:**
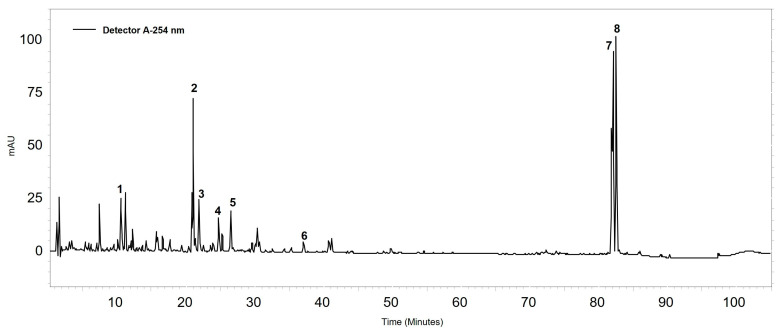
Chromatographic profile of Brazilian red propolis extract at 254 nm, revealing the following compounds: (1) liquiritigenin, (2) formononetin, (3) vestitol, (4) neovestitol, (5) medicarpin, (6) 7-O-neovestitol, (7) guttiferone E, and (8) oblongifolin B. According to Ccana-Ccapatinta, Mejia, Tanimoto, Groppo, Carvalho and Bastos [25].

**Figure 2 antibiotics-13-00719-f002:**
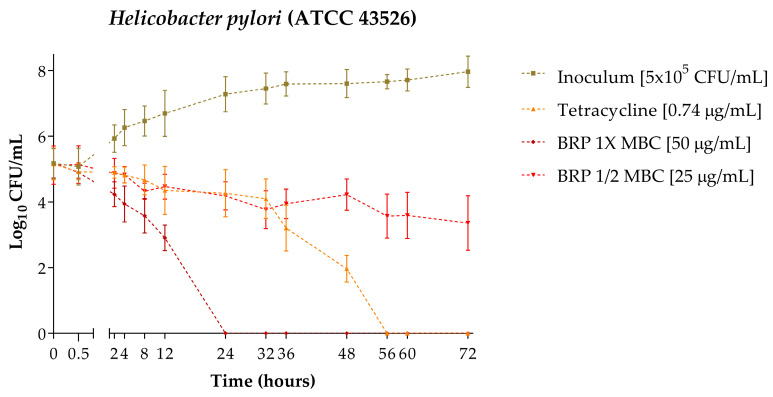
Time–kill curve of *Helicobacter pylori* (ATCC 43526) obtained after 72 h of exposure to the treatments.

**Figure 3 antibiotics-13-00719-f003:**
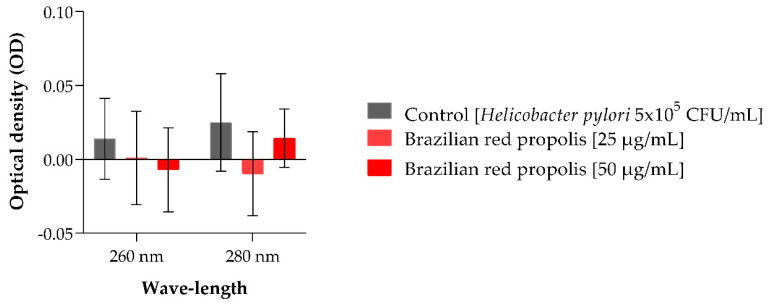
Nucleotide leakage from *Helicobacter pylori* (ATCC 43526) observed after 24 h of exposure to treatments. No statistical differences were found between the treatments and the control.

**Figure 4 antibiotics-13-00719-f004:**
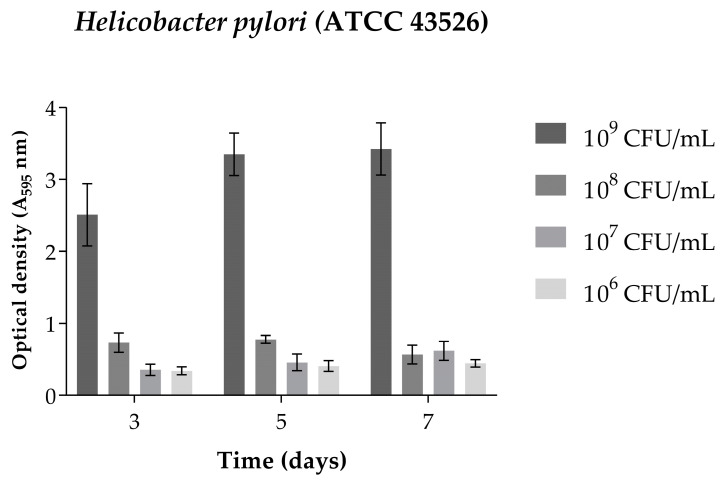
Biofilm formation of *Helicobacter pylori* (ATCC 43526) obtained by optical density (595 nm) after 3, 5, and 7 days of incubation at inoculum concentrations of 10^6^, 10^7^, 10^8^, and 10^9^ CFU/mL.

**Figure 5 antibiotics-13-00719-f005:**
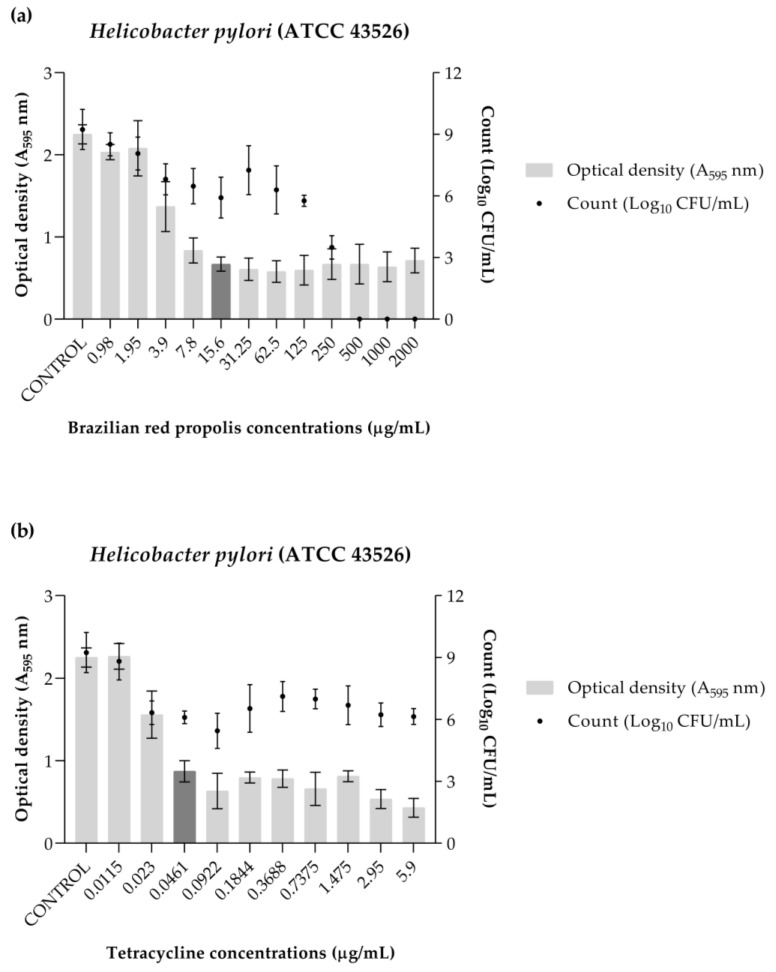
Graphical representation of biofilm formation inhibition of *Helicobacter pylori* (ATCC 43526) as demonstrated by optical density (OD) and numbers of microorganisms (log_10_ CFU/mL). (**a**) Brazilian red propolis. (**b**) Tetracycline. The MICB_50_ value is represented in dark gray.

**Figure 6 antibiotics-13-00719-f006:**
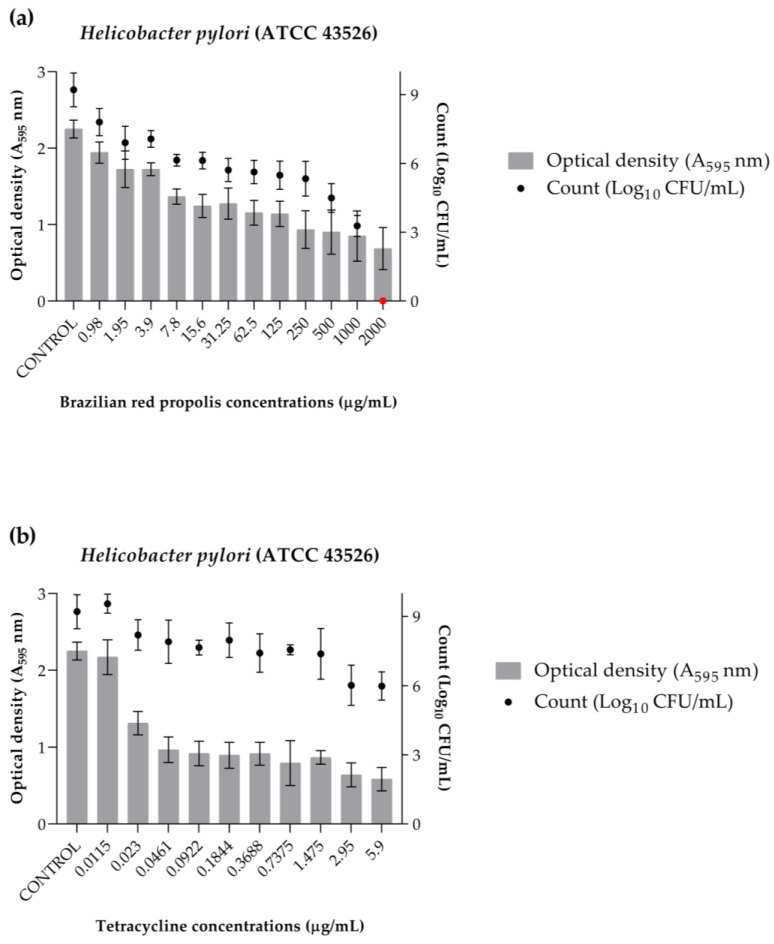
Graphical representation of the eradication of *Helicobacter pylori* (ATCC 43526) biofilm as demonstrated by optical density (OD) and numbers of microorganisms (log_10_ CFU/mL). (**a**) Brazilian red propolis. (**b**) Tetracycline. The MEBC value is represented in red.

## Data Availability

All data generated and analyzed during this study are available from the corresponding author upon reasonable request.

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
