# Peer review of "The Antibacterial Potential of Brazilian Red Propolis against the Formation and Eradication of Biofilm of Helicobacter pylori"

_antibiotics, 2024, doi:10.3390/antibiotics13080719_

Round 1

Reviewer 1 Report

Comments and Suggestions for Authors

1. Figure 1 should be revise the X axis name: Time, minutes

2. Conclusion: add some future plans of projects/applications

3. Why you choose this concentration: 0.98 to 2000 μ g/mL. at 4.2.5 section

4. In 4.1.2 section, do you conduct the quantitative analysis of the detected compounds? what standard of compound did you use?

5. In figure 3, what’s the meaning of negative value in the graphics?

6. Line 100-108 (introduction) still the “instruction to write introduction”, please delete it

Author Response

RESPONSE TO REVIEWER #1

Referee Comment: Figure 1 should be revise the X axis name: Time, minutes.

Answer: We agree with the reviewer and have made the requested change.

Referee Comment: Conclusion: add some future plans of projects/applications.

Answer: We agree with the reviewer and have made the requested change (lines 441-448).

Referee Comment: Why you choose this concentration: 0.98 to 2000 μg/mL. at 4.2.5 section.

Answer: To select the highest concentration evaluated (2000 μg/mL), several concepts were taken into consideration. Firstly, a study from our research group (Santiago et al., 2022) determined the minimum inhibitory concentration (MIC) of Brazilian red propolis against Helicobacter pylori (ATCC 43526) to be 50 μg/mL. Secondly, as discussed in the present manuscript, the dose of an antibacterial agent required to completely eliminate a biofilm can be 10 to 1000 times greater than the dose necessary to eliminate the bacteria in its planktonic form (Mooney et al., 2018). Therefore, the concentration of 2000 μg/mL is 40-fold higher than 50 μg/mL. Our research group has extensive experience regarding the antibiofilm activity of natural products, and promising results are generally found within this concentration range.

References:

Santiago, M.B.; Leandro, L.F.; Rosa, R.B.; Silva, M.V.; Teixeira, S.C.; Servato, J.P.S.; Ambrosio, S.R.; Veneziani, R.C.S.; Aldana-Mejia, J.A.; Bastos, J.K.; et al. Brazilian Red Propolis Presents Promising Anti-H. pylori Activity in In vitro and In vivo Assays with the Ability to Modulate the Immune Response. Molecules 2022, 27, doi:10.3390/molecules27217310.

Mooney, J.A.; Pridgen, E.M.; Manasherob, R.; Suh, G.; Blackwell, H.E.; Barron, A.E.; Bollyky, P.L.; Goodman, S.B.; Amanatullah, D.F. Periprosthetic bacterial biofilm and quorum sensing. J Orthop Res 2018, 36, 2331-2339, doi:10.1002/jor.24019.

Referee Comment: In 4.1.2 section, do you conduct the quantitative analysis of the detected compounds? what standard of compound did you use?

Answer: A validated HPLC-UV method for phenolic compounds in Brazilian red propolis (BRP) was previously developed by our group (Aldana-Mejía et al., 2021). Qualitative analysis was undertaken using this previously developed method. Regarding the polyprenylated benzophenones (guttiferone E and oblongifolin B), which are major compounds in the tested propolis sample, there is no analytical method to quantify these compounds. The internal standard used was benzophenone, purchased from Sigma-Aldrich, and other standards were isolated by our group as described in Ccana-Ccapatinta et al., 2020 and Aldana-Mejía et al., 2021. This study aimed to investigate the antibacterial potential of BRP against Helicobacter pylori. Further studies are necessary to explore which compound is responsible for this biological activity, but the benzophenones might have contributed significantly to the observed activities.

References:

Aldana-Mejía, J. A., Ccana-Ccapatinta, G. V., Ribeiro, V. P., Arruda, C., Veneziani, R. C. S., Ambrósio, S. R., & Bastos, J. K. (2021). A validated HPLC-UV method for the analysis of phenolic compounds in Brazilian red propolis and Dalbergia ecastaphyllum. Journal of pharmaceutical and biomedical analysis, 198, 114029. https://doi.org/10.1016/j.jpba.2021.114029

Ccana-Ccapatinta, G. V., Mejía, J. A. A., Tanimoto, M. H., Groppo, M., Carvalho, J. C. A. S., & Bastos, J. K. (2020). Dalbergia ecastaphyllum (L.) Taub. and Symphonia globulifera L.f.: The Botanical Sources of Isoflavonoids and Benzophenones in Brazilian Red Propolis. Molecules (Basel, Switzerland), 25(9), 2060. https://doi.org/10.3390/molecules25092060

Referee Comment: In figure 3, what’s the meaning of negative value in the graphics?

Answer: The crude extract of Brazilian red propolis has a red coloration. Therefore, it was necessary to subtract the optical density value obtained from the blank (Brazilian red propolis + broth) from the value obtained in the assay (Brazilian red propolis + broth + H. pylori) as described in section 4.2.3: "The absorbance values obtained were subtracted from their respective blanks by subtraction." This mathematical calculation was performed to prevent false results, where the color of the sample could be misinterpreted as nucleotide leakage. Consequently, some values were negative.

Referee Comment: Line 100-108 (introduction) still the “instruction to write introduction”, please delete it.

Answer: The revision has been made.

Reviewer 2 Report

Comments and Suggestions for Authors

1. Why was only one strain of H. pylori used? Testing multiple clinical isolates could offer a more comprehensive understanding of BRP's efficacy.

2. How was the chemical composition of RBP analyzed? The methodology section should thoroughly discuss the detection of compounds. Was there any comparison with standard compounds?. Regarding the compounds detected, authors should at least discuss in the discussion section about its antimicrobial activity, which in this case, lack of information.

3. Since the authors used statistical analysis to determine significant differences between incubation times, this information should be clearly stated in the relevant paragraph. Additionally, please add symbols indicating significant differences in Figure 2 and Figure 4.

4. Based on the methodology, how do the authors ensure that H. pylori produced the biofilm? How do they confirm the biofilm inhibition activity of RBP? The in vitro biofilm model used in the study does not fully replicate the complexity of biofilms in vivo. Have the authors considered using more complex or dynamic models, or conducting in vivo biofilm studies?

5. Paper Structure: Ideally, a paragraph should be 100–200 words long with a maximum of five sentences. This helps readers understand the topic better. Many paragraphs in the article contain only two sentences. Please revise thoroughly to meet these guidelines.

Reviewer 3 Report

Comments and Suggestions for Authors

I would like to extend my congratulations on your work. New studies and contributions on propolis are always welcome by the scientific community. The work is well structured and the methodology is very well described. As you correctly point out, the lack of research on Helicobacter biofilm production does not allow a discussion of the results. However, I feel that the introduction could be elaborated a little more. Thank you very much for your scientific contribution.

Author Response

RESPONSE TO REVIEWER #3

Referee Comment: I would like to extend my congratulations on your work. New studies and contributions on propolis are always welcome by the scientific community. The work is well structured and the methodology is very well described. As you correctly point out, the lack of research on Helicobacter biofilm production does not allow a discussion of the results. However, I feel that the introduction could be elaborated a little more. Thank you very much for your scientific contribution.

Answer: We thank the reviewer for the words of encouragement. We hope that indeed the present study will make a significant scientific contribution. Unfortunately, due to the guidelines of the Journal Antibiotics stating that the introduction section should "briefly place the study in a broad context and highlight why it is important," we are unable to extend it further, as it already occupies one and a half pages in its current state.

Reviewer 4 Report

Comments and Suggestions for Authors

In times of increasing problems of resistance to antibiotics, studying antimicrobial substances of natural origin such as propolis is of great interest. Propolis, already well known for its antimicrobial properties, was the object of this study. The study of propolis from different regions of the globe (red propolis from Brazilian in this case), brings added value as a relevant subject for researchers. The authors highlight in the paper the potential of the crude hydroalcoholic extract of red propolis against the formation and eradication of the biofilm of Helicobacter pylori.

The article was well documented and written, presenting a topic relevant to academia. Although the article is interesting and multidisciplinary, it is not very complex. I would also have expected several red propolis samples from different areas of the country and some simple chemical analyzes for propolis, for example: total content of flavonoids, phenolics etc., but still compensates with the use of HPLC-DAD analysis that separates and characterizes BRP.

The article is good, although there are some errors, omissions in the manuscript. Overall experiments are well done, and the results support the purpose of the article.

Anyway, the study has interesting results that could be published if the writing of this paper and the research are improved. In present form several explanations and modifications are needed.

I have written some suggestions as a way to further improve the study. Below are my specific comments:

Line 2-3: Change the title of the article. Delete the word "Unveiling". Start with: "The Antibacterial Potential of Brazilian Red Propolis Against ….."

Line 35-108: The introduction part is too long. Try to reduce it, especially the first part, the information about H. pylori, which is already known, can be reduced.

Line 100-108: Delete the paragraph from lines 100-108.

Line 100-108: Delete the paragraph from lines 110-112.

Line 109: In general, in the results part, a section does not end with a figure, but with its description. Try a short interpretation of the results according to the figures.

Line 291: Can the authors give details about the year of propolis sampling (from the hives) and its storage conditions until analysis?

Line 291: At materials you mentioned: “Red propolis samples were ....” (in the plural).... Are we talking about a single representative sample or more than one sample? I do not understand.

It would have been interesting if red propolis samples were taken from different regions of Brazil.

For the equipment, reagents(substances) etc. used, it is necessary the manufacturer XXX Company, City, Country. The same way is necessary for used software. Please check the authors' instructions!

The weak points of the study:

The study was carried out on a single reference strain (ATCC 43526). It would have been more interesting if it was done on several strains isolated from patients, possibly resistant to antibiotics.

The potential to eradicate H. pylori biofilm with BRP is primarily due to its complex chemical composition of propolis: flavonoids, phenolic acids and their esters, terpenoids, etc. It would have been interesting to add a more complete characterization of the physical chemical aspects of Brazilian propolis.

Author Response

RESPONSE TO REVIEWER #4

Referee Comment: Line 2-3: Change the title of the article. Delete the word "Unveiling". Start with: "The Antibacterial Potential of Brazilian Red Propolis Against ….."

Answer: We agree with the reviewer and have made the requested change to the title.

Referee Comment: Line 35-108: The introduction part is too long. Try to reduce it, especially the first part, the information about H. pylori, which is already known, can be reduced.

Answer: Taking into consideration that one of the instructions of the journal Antibiotics regarding the Introduction section is that it should be "comprehensible to scientists working outside the topic of the paper," and given that in the current format the introduction of the manuscript is only one and a half pages long, it was decided not to make the suggested alteration made by the reviewer.

Referee Comment: Line 100-108: Delete the paragraph from lines 100-108.

Answer: The revision has been made.

Referee Comment: Line 100-108: Delete the paragraph from lines 110-112.

Answer: The revision has been made.

Referee Comment: Line 109: In general, in the results part, a section does not end with a figure, but with its description. Try a short interpretation of the results according to the figures.

Answer: We thank the reviewer for this observation. Since we divided the Results section into subsections and the journal Antibiotics requires that figures and tables be inserted immediately after their citation in the text, it is necessary to conclude with figures. As starting the citation of a figure in the text requires a brief interpretation of the results, this approach was adopted. We agree that the publisher should use the formatting it deems most suitable for publication.

Referee Comment: Line 291: Can the authors give details about the year of propolis sampling (from the hives) and its storage conditions until analysis?

Answer: The red propolis was acquired in August of 2022, and it was stored in a Freezer at -26 °C. This information has been added to the manuscript (line 287).

Referee Comment: Line 291: At materials you mentioned: “Red propolis samples were ....” (in the plural).... Are we talking about a single representative sample or more than one sample? I do not understand.

Answer: We are talking about a single representative sample. This was corrected in the manuscript (line 286).

Referee Comment: It would have been interesting if red propolis samples were taken from different regions of Brazil.

Answer: It is a good point raised by the referee. In the paper published by Aldana-Mejía et al., (2021), propolis samples were collected from different regions. The composition of different samples varies mainly quantitatively, maintaining a very similar qualitative composition since the botanical sources are the same for all samples. Therefore, the chemical profile of red propolis does not vary significantly, which does not significantly affect its biological activity.

References:

Aldana-Mejía, J. A., Ccana-Ccapatinta, G. V., Ribeiro, V. P., Arruda, C., Veneziani, R. C. S., Ambrósio, S. R., & Bastos, J. K. (2021). A validated HPLC-UV method for the analysis of phenolic compounds in Brazilian red propolis and Dalbergia ecastaphyllum. Journal of pharmaceutical and biomedical analysis, 198, 114029. https://doi.org/10.1016/j.jpba.2021.114029

Referee Comment: For the equipment, reagents(substances) etc. used, it is necessary the manufacturer XXX Company, City, Country. The same way is necessary for used software. Please check the authors' instructions!

Answer: We appreciate the reviewer for the inquiry. A revision of the manuscript was conducted, and the missing information has been added.

Referee Comment: The study was carried out on a single reference strain (ATCC 43526). It would have been more interesting if it was done on several strains isolated from patients, possibly resistant to antibiotics.

Answer: We agree with the reviewer about the inclusion of clinical isolates. In a previous study, our research group evaluated the in vitro and in vivo effects of BRP against the sessile form of two strains of H. pylori (ATCC 43526 and a clinical isolate). The results found by Santiago et al., 2022 were promising and the results of these two strains were not discrepant. Encouraged further studies to better understand this property, unfortunately, during the process of designing the experiment for the present manuscript, technical problems occurred in our laboratory, resulting in the loss of some strains from our culture collection, including the clinical isolate of H. pylori.

Even with this obstacle, we decided to proceed with the study, considering that this is the first work to report the action of BRP against H. pylori. Although reference strains, such as the one used in the present study, may not exhibit the same characteristics as field strains, they remain important because, as reference strains, they allow for the reproducibility of the results obtained (Nysten et al., 2024).

However, although the bacterial strain used in this study being from collection (ATCC 43526), it was isolated from gastric antrum. Therefore, it can be considered an ideal model, taking into account the objective of this work. A screening using different bacteria, including mixed biofilms and clinical isolates, will be performed in the future using the compounds of the Brazilian red propolis.

References:

Santiago, M.B.; Leandro, L.F.; Rosa, R.B.; Silva, M.V.; Teixeira, S.C.; Servato, J.P.S.; Ambrosio, S.R.; Veneziani, R.C.S.; Aldana-Mejia, J.A.; Bastos, J.K.; et al. Brazilian Red Propolis Presents Promising Anti-H. pylori Activity in In vitro and In vivo Assays with the Ability to Modulate the Immune Response. Molecules 2022, 27, doi:10.3390/molecules27217310.

Nysten, J.; Sofras, D.; Van Dijck, P. One species, many faces: The underappreciated importance of strain diversity. PLoS Pathog 2024, 20, e1011931, doi:10.1371/journal.ppat.1011931.

Referee Comment: The potential to eradicate H. pylori biofilm with BRP is primarily due to its complex chemical composition of propolis: flavonoids, phenolic acids and their esters, terpenoids, etc. It would have been interesting to add a more complete characterization of the physical chemical aspects of Brazilian propolis.

Answer: It is a good point raised by the referee. The benzophenones display higher activity compared to other red propolis compounds, contributing greatly to the observed effects. Likewise, the flavonoids and other compounds might enhance these activities, which should be further investigated.

References:

Aldana-Mejía, J. A., Ccana-Ccapatinta, G. V., Ribeiro, V. P., Arruda, C., Veneziani, R. C. S., Ambrósio, S. R., & Bastos, J. K. (2021). A validated HPLC-UV method for the analysis of phenolic compounds in Brazilian red propolis and Dalbergia ecastaphyllum. Journal of pharmaceutical and biomedical analysis, 198, 114029. https://doi.org/10.1016/j.jpba.2021.114029

Ccana-Ccapatinta, G. V., Mejía, J. A. A., Tanimoto, M. H., Groppo, M., Carvalho, J. C. A. S., & Bastos, J. K. (2020). Dalbergia ecastaphyllum (L.) Taub. and Symphonia globulifera L.f.: The Botanical Sources of Isoflavonoids and Benzophenones in Brazilian Red Propolis. Molecules (Basel, Switzerland), 25(9), 2060. https://doi.org/10.3390/molecules25092060

Reviewer 5 Report

Comments and Suggestions for Authors

The findings reported in the paper by Santiago et al. titled “Unveiling the antibacterial potential of Brazilian red propolis against the formation and eradication of biofilm of Helicobacter pylori” are of great practical importance and may contribute to the development of a pharmaceutical formulation effective against Helicobacter pylori. This is particularly important in a situation of developing bacterial resistance to antibiotics currently used to treat the ailment.

A weakness of the paper is the lack of information about the chemical profile of the extract studied. It is true that the authors twice included one and the same sentence (lines 113-115 and 117-119) informing about several secondary metabolites present in the studied extract, but this does not explain anything. Considering the statements made in the Introduction “Due to the seasonality of natural products, the compounds present in propolis can vary according to geographic location or time of collection [16,17] (lines 78-80)” and “Owing to its botanical source and the vast biodiversity of Brazil, Brazilian red propolis (BRP) contains compounds not found in propolis from other sources [21] (lines 86-87)”, it can be concluded with high probability that the study of BRD from another source may not confirm the findings presented in this work. Since it is known that the biological activity of plant material is determined by both the type of secondary metabolites and their concentrations, information on the chemical profile of this particular BRD extract studied is insufficient.

Paragraphs contained in lines 100-108 and 110-112 must be removed from the manuscript.

The same part of the text twice - lines 113-115 and lines 117-119.

Why is the title of the paper in ref. 27 in square brackets?

Ref. 37 - please write the title of the journal correctly.

Author Response

RESPONSE TO REVIEWER #5

Referee Comment: A weakness of the paper is the lack of information about the chemical profile of the extract studied. It is true that the authors twice included one and the same sentence (lines 113-115 and 117-119) informing about several secondary metabolites present in the studied extract, but this does not explain anything. Considering the statements made in the Introduction “Due to the seasonality of natural products, the compounds present in propolis can vary according to geographic location or time of collection [16,17] (lines 78-80)” and “Owing to its botanical source and the vast biodiversity of Brazil, Brazilian red propolis (BRP) contains compounds not found in propolis from other sources [21] (lines 86-87)”, it can be concluded with high probability that the study of BRD from another source may not confirm the findings presented in this work. Since it is known that the biological activity of plant material is determined by both the type of secondary metabolites and their concentrations, information on the chemical profile of this particular BRD extract studied is insufficient.

Answer: We are grateful for the comment. Regarding lines 78-80 and 86-87, the biological activity of propolis in general indeed depends on its botanical source, which determines the type of secondary metabolites and the corresponding biological activities observed. In the introduction, we mention that the composition of propolis varies mainly based on its botanical sources, which differ by region. In Brazil, there are several different types of propolis, as reported by Salatino et al., 2021. Both red and green propolis are chemically and botanically well characterized, allowing for the production of standardized extracts.

As cited in lines 106-107 and 302, the chemical profile of Brazilian red propolis (BRP) extract is already known and has been extensively studied since 2007 (Alencar et al., 2007). Our group has published relevant results on the botanical sources and seasonality of BRP, as reported by Aldana-Mejía et al., 2021, and Ccana-Ccapatinta et al., 2020, respectively.

References:

Aldana-Mejía, J. A., Ccana-Ccapatinta, G. V., Ribeiro, V. P., Arruda, C., Veneziani, R. C. S., Ambrósio, S. R., & Bastos, J. K. (2021). A validated HPLC-UV method for the analysis of phenolic compounds in Brazilian red propolis and Dalbergia ecastaphyllum. Journal of pharmaceutical and biomedical analysis, 198, 114029. https://doi.org/10.1016/j.jpba.2021.114029

Alencar, S. M., Oldoni, T. L., Castro, M. L., Cabral, I. S., Costa-Neto, C. M., Cury, J. A., Rosalen, P. L., & Ikegaki, M. (2007). Chemical composition and biological activity of a new type of Brazilian propolis: red propolis. Journal of ethnopharmacology, 113(2), 278–283. https://doi.org/10.1016/j.jep.2007.06.005

Ccana-Ccapatinta, G. V., Mejía, J. A. A., Tanimoto, M. H., Groppo, M., Carvalho, J. C. A. S., & Bastos, J. K. (2020). Dalbergia ecastaphyllum (L.) Taub. and Symphonia globulifera L.f.: The Botanical Sources of Isoflavonoids and Benzophenones in Brazilian Red Propolis. Molecules (Basel, Switzerland), 25(9), 2060. https://doi.org/10.3390/molecules25092060

Salatino, A., Salatino, M. L. F., & Negri, G. (2021). How diverse is the chemistry and plant origin of Brazilian propolis?. Apidologie, 52(6), 1075–1097. https://doi.org/10.1007/s13592-021-00889-z 

Referee Comment: Paragraphs contained in lines 100-108 and 110-112 must be removed from the manuscript.

Answer: The revision has been made.

Referee Comment: The same part of the text twice - lines 113-115 and lines 117-119.

Answer: The revision has been made.

Referee Comment: Why is the title of the paper in ref. 27 in square brackets?

Answer: The revision has been made.

Referee Comment: Ref. 37 - please write the title of the journal correctly.

Answer: The revision has been made.

Round 2

Reviewer 4 Report

Comments and Suggestions for Authors

The authors answered some of the questions, but they did not really make all the changes in the text.

I still maintain that the Introduction can be shortened, because in the Instructions for authors it is said that: “briefly place the study in a broad context”. Even if it is shorter and more concise, the Introduction can be “comprehensible to scientists working outside the topic of the paper”.

About the figures, each section of the results can start with a phrase like: “The results obtained in …….. were represented in Figure ….”, then the figure is placed, after which it is described.

The answer regarding why samples were not taken from different regions of Brazil should be entered in the text, in the Discussion or in the limitations of the study.

The answer given regarding only one reference strain was used should be entered in the text, in the Discussion or in the limitations of the study.

Author Response

RESPONSE TO REVIEWER #4 – Round 2

Referee Comment: I still maintain that the Introduction can be shortened, because in the Instructions for authors it is said that: “briefly place the study in a broad context”. Even if it is shorter and more concise, the Introduction can be “comprehensible to scientists working outside the topic of the paper”.

Answer: Changes were made to summarize the content (lines 35-45; 48-51; 55-60; 62-64).

Referee Comment: About the figures, each section of the results can start with a phrase like: “The results obtained in …….. were represented in Figure ….”, then the figure is placed, after which it is described.

Answer: We appreciate the reviewer's suggestion; however, we have decided to retain the current formatting of the Results section. This decision is based on the fact that the journal Antibiotics has recently published other articles (Artesani et al., 2024; Campanimi-Salinas et al., 2024; Jeon et al., 2024; Mussin & Giusiano, 2024) with the same formatting style (e.g., results, subsections, description of results, figure/table). However, we still agree that the publisher should use the formatting it deems most suitable for publication.

References:

Artesani, L.; Ciociola, T.; Vismarra, A.; Bacci, C.; Conti, S.; Giovati, L. Activity of Synthetic Peptide KP and Its Derivatives against Biofilm-Producing Escherichia coli Strains Resistant to Cephalosporins. Antibiotics 2024, 13, doi:10.3390/antibiotics13080683.

Campanini-Salinas, J.; Opitz-Ríos, C.; Sagredo-Mella, J.A.; Contreras-Sanchez, D.; Giménez, M.; Páez, P.; Tarifa, M.C.; Rubio, N.D.; Medina, D.A. Antimicrobial Resistance Elements in Coastal Water of Llanquihue Lake, Chile. Antibiotics 2024, 13, doi:10.3390/antibiotics13070679.

Jeon, Y.-N.; Ryu, S.-J.; Lee, H.-Y.; Kim, J.-O.; Baek, J.-S. Green Synthesis of Silver Nanoparticle Using Black Mulberry and Characterization, Phytochemical, and Bioactivity. Antibiotics 2024, 13, doi:10.3390/antibiotics13080686.

Mussin, J.; Giusiano, G. Synergistic Antimicrobial Activity of Biogenic Silver Nanoparticles and Acanthospermum australe Essential Oil against Skin Infection Pathogens. Antibiotics 2024, 13, doi:10.3390/antibiotics13070674.

Referee Comment: The answer regarding why samples were not taken from different regions of Brazil should be entered in the text, in the Discussion or in the limitations of the study.

Answer: The change suggested by the reviewer has been made (lines 268-281).

Referee Comment: The answer given regarding only one reference strain was used should be entered in the text, in the Discussion or in the limitations of the study.

Answer: The change suggested by the reviewer has been made (lines 282-288).

Reviewer 5 Report

Comments and Suggestions for Authors

The authors did not correctly explain the lack of information on the chemical profile of the extracts studied. They only indicated that the relevant data can be found in the cited publications. However, I believe that writing in the “Introduction” section about the very high biodiversity of Brazilian red propolis, at least a short paragraph discussing this issue should be included in the “Introduction”. This is very important, since the biological activity of a given extract is determined precisely by the secondary metabolites they contain. The inclusion of a short paragraph would enable the reader to better understand the manuscript without having to delve into the content of the cited publications.

Author Response

RESPONSE TO REVIEWER #5 – Round 2

Referee Comment: The authors did not correctly explain the lack of information on the chemical profile of the extracts studied. They only indicated that the relevant data can be found in the cited publications. However, I believe that writing in the “Introduction” section about the very high biodiversity of Brazilian red propolis, at least a short paragraph discussing this issue should be included in the “Introduction”. This is very important, since the biological activity of a given extract is determined precisely by the secondary metabolites they contain. The inclusion of a short paragraph would enable the reader to better understand the manuscript without having to delve into the content of the cited publications.

Answer: We implemented the reviewer's suggestion and added information regarding the chemical profile of BRP to the Introduction (lines 72-79). Additionally, a paragraph discussing the compounds identified in BRP had been included in the Discussion section of the manuscript (lines 256-281). We hope these changes make the manuscript more comprehensive regarding the compounds found in BRP.